# Energy-Based Constraint Networks:
# Learning Structural Coherence Across Modalities

## Abstract

We introduce energy-based constraint networks—a modality-agnostic architecture that learns structural coherence from contrastive pairs. The system processes frozen encoder embeddings through a state-space model with dual-head attention, producing a scalar energy measuring structural consistency alongside per-position energy scores that localize violations. Multiple independently trained branches detect different violation types and compose at inference without interference.

We demonstrate the framework in two domains. In text, the system achieves 93.4% accuracy on trained corruption types and 87.2% on 9 unseen types, using frozen BERT and 7.4M trainable parameters. In vision, the same architecture achieves competitive deepfake detection: 0.959 AUC on FaceForensics++ Deepfakes and 0.870 on Celeb-DF without any Celeb-DF training data, using frozen DINOv2 and 3.6M parameters per branch.

The framework supports flexible training: branches learn from designer-specified corruptions, real-world paired data, or both. Composable branches require representation compatibility—a finding validated through extensive experimentation where five incompatible approaches failed before the compatible one succeeded. The architecture is encoder-agnostic and domain-agnostic: changing the domain requires only new corruption strategies; changing the encoder requires only a new input projection layer. To our knowledge, this is the first architecture to learn within-modality structural coherence as an explicit energy landscape with per-position decomposition, and to demonstrate that the same architecture applies across modalities via corruption respecification alone—without sharing any learned parameters between domains.

## 1 Introduction

### 1.1 Structural Coherence as a Learnable Property

Structural coherence—the property that parts of a well-formed artifact are mutually consistent—spans domains. A coherent paragraph maintains consistent tense, reference, and topic across sentences. A coherent face image maintains consistent lighting, texture, and geometry across spatial regions. In both cases, violations of coherence indicate manipulation, whether corrupted text or deepfake images.

We propose that structural coherence is a distinct, learnable distributional property. Autoregressive language models learn $P(\text{next token} \mid \text{context})$—a local conditional distribution. They can assign high probability to every individual token in a self-contradicting paragraph, because each token is locally plausible. Energy-based approaches to text (Deng et al., 2020) and discourse coherence models (Barzilay & Lapata, 2008; Li & Hovy, 2014) have explored global sequence-level evaluation, but typically operate without per-position decomposition, require task-specific supervision, or are limited to a single modality. Our system learns structural coherence as an explicit energy landscape with per-position violation localization, specified through corruption examples rather than labels—and the same architecture applies across modalities by respecifying the corruption strategy alone.

This distinction extends beyond text. Deepfake generators produce images where each pixel is locally plausible, but global structural properties (lighting consistency, texture uniformity, frequency profiles) are violated. The same energy-based framework that detects structural violations in text detects them in images, because the underlying concept—global coherence across positions—is modality-independent.

## 1.2 Corruption as Specification

The central design principle is that structural properties are specified implicitly through corruptions that violate them. Shuffling sentences teaches discourse ordering. Splicing textures teaches regional compatibility. The corruption strategy *is* the specification language: each corruption type encodes a structural property, and the model learns to check it.

This framing has a practical consequence: new structural constraints are added by designing new corruptions, without modifying the model, the training procedure, or the loss function. We validate this empirically: the text model, trained on 6 corruption types, generalizes to 87.2% accuracy on 9 unseen corruption types. The vision model, trained on synthetic corruptions alone, achieves 0.850 AUC on deepfakes it was never trained on. In both cases, the model has learned structural coherence as a general property, not the specific corruption patterns in its training data.

## 1.3 Contributions

1. **Corruption-as-specification as a modality-general framework**: The same architecture learns structural coherence in both text and vision, achieving 87.2% generalization to unseen text corruptions and 0.850 AUC zero-shot deepfake detection from corruption training alone.

2. **Per-position violation localization**: The energy function decomposes per position, localizing where violations occur. In text, per-window energies show how inconsistency propagates through SSM state. In vision, per-patch energies reshape to a $16{\times}16$ spatial heatmap localizing manipulated regions.

3. **Representation compatibility for composable constraints**: Composing constraint branches requires representation compatibility—a finding validated by five failed approaches where different feature spaces produced incompatible energy scales. Processing both views through the same frozen encoder achieves 44.4% frequency contribution, compared to less than 3% for all incompatible approaches. The resulting multi-branch system provides robustness across unknown manipulation types: no single branch covers all methods, but the combination handles diverse manipulation types without requiring knowledge of which type is present at test time.

4. **Flexible training from corruptions, real data, or both**: The system operates across a spectrum from zero-shot detection (corruption specifications only) to supervised detection (paired real/fake data), with corruption pretraining consistently improving cross-dataset transfer.

5. **Independent specialist branches**: Independently trained branches avoid the interference documented in five failed joint-training approaches (Appendix A), enabling modular addition of new constraint types without retraining existing ones. The combined system provides robustness across manipulation types rather than uniformly outperforming every individual branch on every method.

# 2 Architecture

## 2.1 Constraint Network

Each constraint branch takes a sequence of embedding vectors—$(\text{batch}, \text{positions}, \text{dim})$—and produces a scalar energy plus per-position energy decomposition. The architecture is identical across branches and modalities.

**SSM Backbone** (6 blocks): Depthwise causal convolution with gated linear units and learned exponential decay. Captures local sequential patterns and state dynamics across the sequence at linear computational cost.

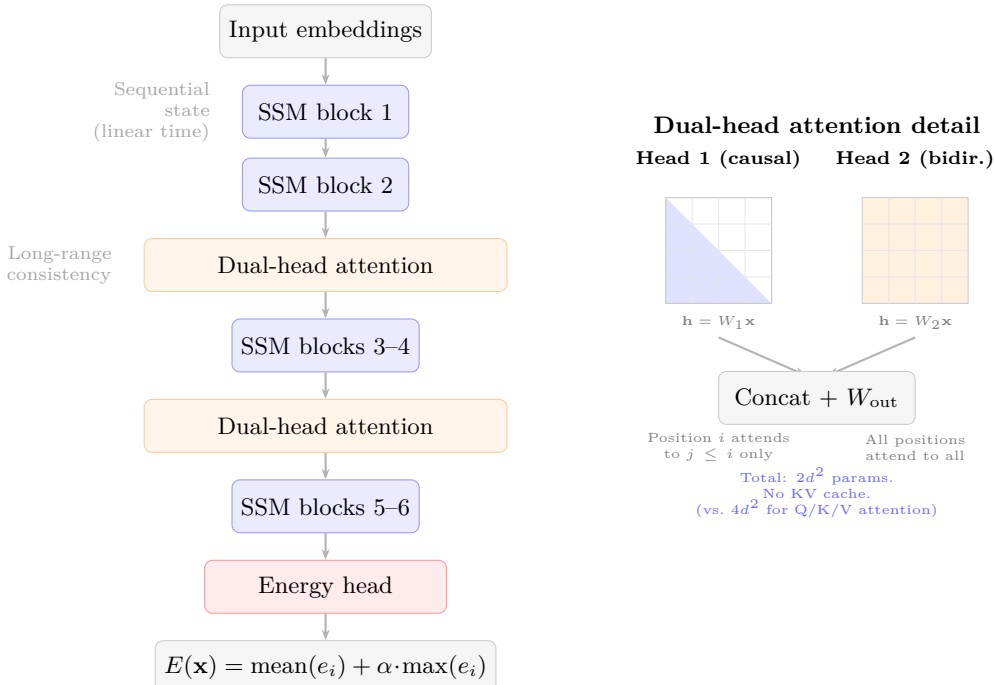

Figure 1: Left: constraint network architecture, shared across modalities and branches. SSM blocks handle sequential state tracking at linear cost; dual-head attention blocks perform long-range consistency checks. Right: the dual-head attention mechanism. Each head uses a single learned projection $W$ (not separate Q/K/V), reducing attention parameters from $4d^2$ to $2d^2$ and eliminating key-value cache entirely—each evaluation is a stateless forward pass. Asymmetry emerges from the masking structure, not from parameterization.

**Dual-Head Attention** (2 blocks, interleaved at positions 2 and 4): Head 1 uses causal masking (forward consistency); Head 2 is bidirectional (mutual compatibility). Each head uses a single learned projection $W_i \in \mathbb{R}^{d \times d/2}$ (not separate Q/K/V). Standard multi-head attention requires $W_Q, W_K, W_V \in \mathbb{R}^{d \times d}$ plus $W_O \in \mathbb{R}^{d \times d}$, totaling $4d^2$ parameters. Our design uses $W_1 \in \mathbb{R}^{d \times d/2}$, $W_2 \in \mathbb{R}^{d \times d/2}$, and $W_O \in \mathbb{R}^{d \times d}$, totaling $2d^2$ parameters—a 50% reduction. The tradeoff is expressiveness: a single projection per head cannot independently model asymmetric attention patterns (where the relevance of position $j$ to position $i$ differs from $i$ to $j$). Our design recovers asymmetry through masking structure—Head 1's causal mask breaks symmetry directionally—rather than through parameterization. Because there are no separate K and V matrices, there is no key-value cache to store or grow—each forward pass is stateless, with the SSM's fixed-size recurrent state as the only persistent memory. This makes the constraint network lightweight to evaluate repeatedly during training or as a streaming coherence monitor at inference. Figure 1 shows the full architecture.

**Energy Head**: LayerNorm $\rightarrow$ MLP $\rightarrow$ per-position energies $e_i$. Aggregation:

$$E(\mathbf{x}) = \text{mean}(e_i) + \alpha \cdot \text{max}(e_i), \quad \alpha = 0.3 \tag{1}$$

The max term addresses a structural property of localized violations: a face-swap boundary affects roughly 10 of 256 patches, and a topic splice affects 2 of ~20 windows, where pure mean aggregation risks diluting the signal. However, ablation (Appendix E) shows that the max term helps equally for sparse violations (face swaps) and diffuse violations (whole-face reenactment), indicating it provides a complementary worst-case signal rather than specifically rescuing sparse cases. The mean term captures aggregate incoherence; the max term captures peak violation intensity; both contribute.

The coefficient $\alpha = 0.3$ was selected in the text domain and carried across to vision without retuning. Ablation confirms the system is insensitive to $\alpha$ in $[0.2, 1.0]$: any value in this range produces comparable

results across all benchmarks (Appendix E). The cross-modal applicability of a single hyperparameter is consistent with the architectural claim that both domains perform the same kind of structural reasoning.

**Contrastive Loss**:

$$\mathcal{L} = E(\mathbf{x}_{\mathrm{pos}})^2 + \max(0, m - E(\mathbf{x}_{\mathrm{neg}}))^2 \tag{2}$$

where $m = 5.0$, pushing coherent inputs toward zero energy and violations above the margin.

## 2.2 Violation Propagation Through SSM State

A key property of the SSM backbone is that violations propagate beyond their local position. The SSM's recurrent state carries information forward, so a violation at one position corrupts the state for all subsequent positions. In text, a topic splice raises energy not only at the spliced sentences but at all surrounding sentences whose contextual relationships have been disrupted. In vision, a face swap boundary elevates energy at adjacent patches whose contextual relationships are inconsistent.

## 2.3 Composable Multi-Branch System

The full vision system composes independently trained branches, each specializing in different violation types (Figure 2):

**Branch 1—Structural**: Detects global structural violations (face swaps, expression transfers, identity manipulations). Processes encoder features from the original input.

**Branch 2—Frequency**: Detects frequency-domain anomalies (GAN smoothing, texture loss). Processes encoder features from a frequency heatmap rendering—the same frozen encoder applied to a different view of the same input.

**Branch 3—Local Texture**: Detects localized texture inconsistencies (neural rendering, region-level quality differences). Processes encoder features from the original input with independently trained weights.

Composition:

$$E = E_{\mathrm{structural}} + \mathrm{gate} \cdot E_{\mathrm{frequency}} + \beta \cdot E_{\mathrm{local}}, \quad \beta = 0.3 \tag{3}$$

where the frequency gate is a heuristic z-score with no learned parameters: baseline frequency statistics (median and standard deviation of the spectral ratio) are computed from real images in the target domain, and each image's gate value is the clipped z-score of its frequency ratio relative to this baseline (gate $= \mathrm{clip}(z/2, 0, 1)$). A dataset-level threshold disables the frequency branch entirely when real images show normal frequency profiles (mean gate $> 0.20$), preventing the near-chance frequency branch from degrading the combined score. Baseline statistics use only real images—no fake images or labels—analogous to computing normalization statistics in anomaly detection. Each branch is frozen at inference.

## 2.4 Encoders

The constraint network operates on embeddings from a frozen encoder (Figure 3). The encoder determines the quality ceiling—a principle validated in both modalities.

**Text**: BERT-base-uncased (110M parameters, frozen). Hidden states from layer $-2$, pooled into 8-token windows, producing $(\mathrm{num\_windows}, 768)$. BERT embedding distance between original and shuffled text is 12.16; MiniLM gives approximately zero—directly explaining why BERT achieves 93.4% while MiniLM achieves 54.5%.

**Vision**: DINOv2 ViT-B/14 (86M parameters, frozen). $16 \times 16 = 256$ patch tokens of 768 dimensions. DINOv2's self-supervised training preserves spatial structure. A corruption alignment diagnostic confirmed DINOv2 encodes structural features well but is blind to frequency-domain features—motivating the multi-branch architecture.

A single learned linear projection ($768 \rightarrow 384$) adapts encoder outputs to the constraint network. This is the only component that changes between modalities.

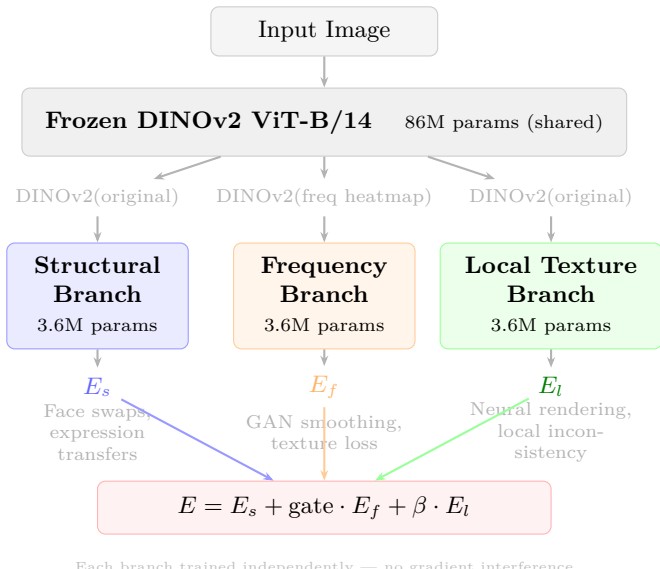

Figure 2: Three-branch composable architecture for vision. All branches process features from the same frozen DINOv2 encoder, ensuring representation compatibility. The frequency branch encodes a rendered frequency heatmap through DINOv2 rather than raw frequency features—the key insight from the compatibility investigation (Section 6). Each branch trains independently; composition occurs only at inference.

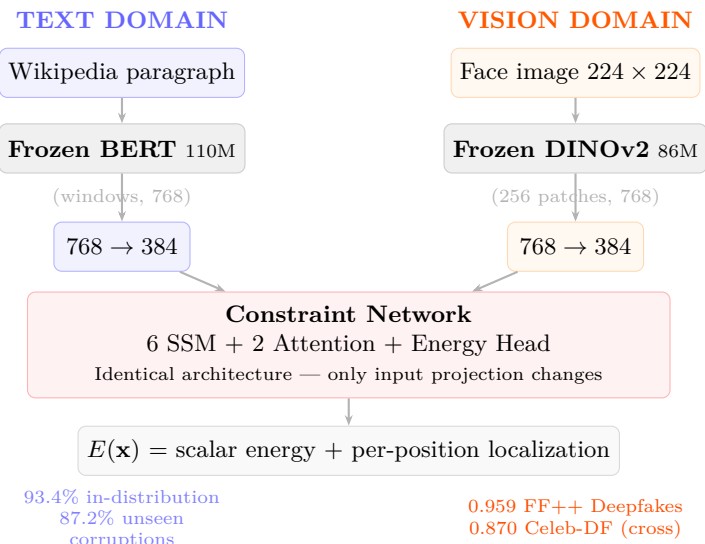

Figure 3: Cross-modal applicability: the same constraint network architecture processes both text (BERT windows) and vision (DINOv2 patches). Only the input projection layer changes between modalities. The architecture, loss function, and training procedure are identical. No learned parameters are shared between domains.

## 3 Training

### 3.1 Corruption-as-Specification

Branches learn from designer-specified corruptions applied to real data. Each corruption implicitly defines a structural property:

**Text corruptions**: Shuffle (discourse ordering), negate (logical consistency), entity swap (referential integrity), tense shift (temporal consistency), coreference break (identity consistency), topic splice (thematic coherence).

**Vision corruptions**: Texture splice (regional compatibility), lighting shift (illumination consistency), localized smoothing (texture quality uniformity), bilateral filtering (edge-preserving blur detection).

Critically, the vision corruption set was derived as a theoretical specification of facial coherence—answering "what makes a face structurally consistent?" rather than "what does a deepfake look like?" The corruptions were designed without reference to any deepfake method's artifact profile. That the resulting detector catches face swaps is not because the corruptions were engineered to match face-swap artifacts; it is because face swaps are, by definition, a violation of regional compatibility, and texture splice is what regional compatibility violation looks like when specified as a corruption.

### 3.2 Learning from Real-World Data

Branches also learn directly from paired real/manipulated data. In vision, matched real/fake pairs from FaceForensics++ (Rössler et al., 2019) train the structural branch to detect deepfake-specific artifacts that synthetic corruptions may not fully capture.

### 3.3 Combining Both

The strongest results come from combining corruption pretraining with real-data fine-tuning. Corruption pretraining establishes what "coherent" looks like broadly. Paired fine-tuning adds method-specific precision. The local texture branch trains on synthetic localized corruptions *and* real NeuralTextures manipulations, achieving L-AUC 0.911 on NeuralTextures and L-AUC 0.871 on the entirely unseen Celeb-DF (Li et al., 2020b) dataset.

### 3.4 Independent Training

Each branch trains in complete isolation. No gradients flow between branches. This eliminates interference: including texture corruptions in the structural branch improved NeuralTextures ($0.800 \rightarrow 0.843$) but degraded cross-dataset Celeb-DF transfer ($0.820 \rightarrow 0.787$). The corruption types that help one task actively harm another. Independent training with composition at inference resolves this completely.

All embeddings are pre-computed and cached (float16), making training epochs pure tensor operations—approximately 30 minutes per branch on a single GPU. This tractability is by design: the framework separates the expensive computation (frozen encoder, run once) from the trainable computation (constraint network, 3.6–7.4M parameters), enabling full reproduction of all experiments in this paper on a single consumer GPU in under a day.

## 4 Results

### 4.1 Text Domain

50,000 Wikipedia paragraphs (Merity et al., 2017), 6 corruption types, frozen BERT encoder, 7.4M trainable parameters. Results are stable across random initializations: overall validation accuracy $0.788 \pm 0.002$ over 5 seeds (min 0.785, max 0.790). Per-type standard deviations range from 0.002 (tense shift, negate) to 0.010 (shuffle).

The generalization result (Table 1) is the key validation of corruption-as-specification. Accuracy is computed on valid pairs only—corruptions that actually modified the text—since silent failures (where the corruption function returns unchanged text) produce identical positive/negative pairs that are uninformative. The Skip column in Table 1 shows that tense shift fails silently 48% of the time and coreference break 32%, while all unseen corruption types succeed on $\geq 98\%$ of paragraphs. The model detects role/title swaps at 100%, demonstrative confusion at 96%, and bridging reference breaks at 93%—none of which appeared in training.

Table 1: Text domain: detection accuracy on all 15 corruption types. The model was trained on only the first 6. The 9 below the line were never seen during training. "Valid" = pairs where the corruption actually modified the text; "Skip" = silent failures where the corruption returned unchanged text (filtered from accuracy computation). Out of 100 paragraphs per type.

| Corruption | Structural property | Acc. | Gap | Valid | Skip |
|---|---|---|---|---|---|
| *Trained corruption types* | | | | | |
| Tense shift | Temporal consistency | 100% | +0.57 | 52 | 48 |
| Coref. break | Identity consistency | 100% | +0.46 | 68 | 32 |
| Negate | Contradiction | 99% | +0.33 | 99 | 1 |
| Topic splice | Topical coherence | 98% | +0.69 | 100 | 0 |
| Entity swap | Referential integrity | 90% | +0.15 | 91 | 9 |
| Shuffle | Discourse ordering | 87% | +0.55 | 100 | 0 |
| *Unseen corruption types (never in training data)* | | | | | |
| Role/title swap | Role-entity binding | 100% | +0.33 | 100 | 0 |
| Demonstrative | Demonstrative reference | 96% | +0.22 | 100 | 0 |
| Description swap | Definite description | 93% | +0.15 | 100 | 0 |
| Bridging break | Bridging inference | 93% | +0.17 | 100 | 0 |
| Singular/plural | Number agreement | 93% | +0.15 | 99 | 1 |
| Name substitution | Person identity | 90% | +0.11 | 100 | 0 |
| Temporal break | Temporal consistency | 88% | +0.09 | 100 | 0 |
| Number change | Numerical consistency | 77% | +0.15 | 98 | 2 |
| Repetition | Discourse progression | 44% | −0.01 | 100 | 0 |
| **Trained (6 types)** | | **93.4%** | | 510 | |
| **Unseen (9 types)** | | **87.2%** | | 897 | |
| **Overall (15 types)** | | **89.3%** | | 1407 | |

The one genuine failure is repetition (44%): verbatim repetition is locally coherent at the representation level, a limitation of any approach operating on embeddings rather than raw tokens.

The encoder comparison confirms that the encoder determines the quality ceiling:

Table 2: Text domain: encoder comparison at the same training data scale (50K paragraphs).

| Corruption | MiniLM 2K | MiniLM 50K | BERT 50K | Feature |
|---|---|---|---|---|
| Shuffle | 45% | 67% | 84% | Word order |
| Entity swap | 42% | 48% | 74% | Entity names |
| Negate | 95% | 82% | 88% | Semantics |
| Topic splice | 60% | 60% | 74% | Topic flow |
| Coref. break | 53% | 55% | 65% | Pronouns |
| Tense shift | 32% | 56% | 59% | Morphology |
| **Overall** | 54.5% | 62.8% | 78.7% | |

## 4.2 Structural Coherence in Generated Text

The corruption detection results above evaluate whether the constraint network can distinguish clean text from synthetically corrupted text. A stronger test: can it detect structural violations in text generated by a language model—real hallucinations, not synthetic corruptions?

We generated 200 paragraphs from GPT-2-medium (355M parameters) and scored each with both perplexity and constraint energy. A human annotator labeled 30 high-energy and 30 low-energy paragraphs for structural coherence (entity consistency, referential integrity, temporal order, topical continuity), blind to the energy scores. Paragraphs containing controversial content were excluded and replaced. No standard benchmark exists for evaluating structural coherence detection independently of distributional familiarity; this evaluation was constructed for the purpose.

Table 3: Detection of structural violations in GPT-2-medium generated text. The constraint network (7.4M params) separates coherent from broken generated text substantially better than perplexity (355M params). Human labels: 1 = structurally coherent, 0 = broken (entity confusion, contradictions, referential failures, topic drift).

| Metric | Coherent (n=35) | Broken (n=27) | Gap | AUC | p-value |
|---|---|---|---|---|---|
| Constraint energy | $+2.62 \pm 0.57$ | $+4.28 \pm 0.41$ | $+1.66$ | **0.945** | $< 10^{-6}$ |
| Perplexity | $28.0 \pm 9.1$ | $36.3 \pm 6.2$ | $+8.4$ | 0.803 | 0.0002 |

The constraint network achieves 0.945 AUC: of the 30 highest-energy paragraphs, 27 were labeled broken; of the 30 lowest-energy, all 30 were labeled coherent. Perplexity shows partial separation (AUC 0.803) but this reflects co-occurrence of surface-level token disfluency with structural violations in GPT-2-medium's output, not structural coherence detection per se—consistent with the well-established finding that perplexity thresholding does not address hallucination in language models.

This is the strongest generalization result in the paper. The constraint network was trained on synthetic corruptions of Wikipedia text. It has never seen GPT-2's output or real hallucinations. Yet it separates structurally coherent from structurally broken generated text at 0.945 AUC—providing direct evidence that corruption-as-specification trains the model to detect structural coherence as a general property, not the specific corruption patterns in its training data. We note the limitations of this pilot evaluation: a single annotator, 62 labeled paragraphs sampled from the energy extremes, and one generator (GPT-2-medium). The effect size is large ($t = 12.6$, $p < 10^{-6}$) but validation with multiple annotators, larger samples, and diverse generators would strengthen the finding.

### 4.3 Vision Domain

The three-branch system on FaceForensics++ (Rössler et al., 2019) (5 methods) and Celeb-DF (Li et al., 2020b) (cross-dataset, zero training data from this set):

Table 4: Vision domain: deepfake detection AUC (mean ± std over 5 seeds). Celeb-DF is cross-dataset (no training data from this set).

| Method | Structural | Frequency | Local | Combined |
|---|---|---|---|---|
| FF++ Deepfakes | 0.959 | 0.636 | 0.834 | **0.959±.001** |
| FF++ Face2Face | — | — | — | 0.909±.003 |
| FF++ FaceSwap | — | — | — | 0.919±.003 |
| FF++ NeuralTextures | 0.819 | 0.509 | 0.911 | **0.880±.005** |
| FF++ FaceShifter | — | — | — | 0.897±.005 |
| **Celeb-DF (cross)** | 0.820 | 0.561 | 0.871 | **0.870±.019** |

Results are stable across random initializations: FF++ methods show standard deviations of 0.001–0.005 over 5 seeds. Cross-dataset transfer to Celeb-DF shows wider variance (±0.019), expected when generalizing to an unseen distribution where small weight differences at the decision boundary matter more.

Table 5: Effect of training data source on detection performance.

| Training Data | FF++ Deepfakes | FF++ NT | Celeb-DF |
|---|---|---|---|
| Corruptions only (zero deepfakes) | 0.850 | 0.489 | 0.698 |
| Real deepfake pairs only | 0.963 | 0.800 | 0.820 |
| Combined (three-branch) | 0.959 | 0.880 | **0.870** |

### 4.4 Training Flexibility

With zero deepfake training data, the model detects face swaps at 0.850 AUC (Table 5)—purely from learning what structurally coherent faces look like. The texture_splice corruption aligns with face swap boundary artifacts, just as text shuffling aligns with discourse ordering violations. Adding real paired data and specialist branches pushes cross-dataset transfer from 0.698 to 0.870.

### 4.5 Comparison with Published Methods

Table 6: Cross-dataset evaluation: trained on FF++, tested on Celeb-DF. Single-frame detection unless noted. † = frozen/minimal encoder adaptation.

*Effort: average frame-level AUC on Celeb-DF++ intra-scenario; most directly comparable single-frame protocol.

**GenD: video-level AUROC averaging 32 frames per video; averaging smooths per-frame noise, making this a substantially easier metric than single-frame AUC.

| Method | Year | Params | Celeb-DF AUC | Training |
|---|---|---|---|---|
| Face X-ray | 2020 | ∼25M | 0.742 | End-to-end |
| GFF | 2021 | ∼30M | 0.794 | End-to-end |
| LTW | 2021 | ∼25M | 0.771 | End-to-end |
| RECCE | 2022 | ∼30M | 0.687 | End-to-end |
| CAST-B0 | 2025 | ∼25M | 0.770 | End-to-end |
| GA-LASSO+SVM | 2026 | ∼25M | 0.787 | Hybrid |
| Effort† (ICML) | 2025 | ∼300M | 0.844* | LN tuning |
| GenD† | 2025 | ∼304M | 0.882** | LN tuning |
| SDEQ-Net (video) | 2026 | ∼40M | 0.896 | End-to-end, multi |
| **Ours (3-branch)†** | **2026** | **10.8M** | **0.870±.019** | **Frozen DINOv2** |

The system achieves 0.870 frame-level AUC on Celeb-DF (mean over 5 seeds). Recent foundation-model approaches (GenD, Effort) report strong cross-dataset video-level AUROC using ViT-L-scale encoders (∼300M parameters) with LayerNorm or subspace adaptation; our system operates at frame level with a smaller encoder (86M frozen DINOv2) and 10.8M trainable parameters total, while supporting cross-modal applicability that encoder-adaptation methods cannot (Table 6). GenD's 0.882 is video-level AUROC with 32-frame averaging—a substantially easier task than single-frame detection, as averaging smooths per-frame noise and lets weak signals accumulate. Without their frame-level number, a direct comparison is not possible. The only method substantially exceeding our result (SDEQ-Net at 0.896) also uses multi-frame video analysis.

## 5 What the Model Learns

### 5.1 Structure, Not Content

The constraint network learns distributional properties of how positions relate to each other in embedding space, not the content at any individual position. In text, a sentence about nuclear physics and a sentence about basketball have different embeddings, but the model cares about whether the *transition* follows patterns learned from coherent text. In vision, a patch showing skin and a patch showing hair have different

features, but the model cares about whether the texture *transition* between them is consistent with natural faces.

## 5.2 Why Generalization Works

The 87.2% accuracy on unseen text corruptions and 0.850 AUC on unseen deepfakes require explanation. The model was never trained on role/title swaps, yet detects them at 100%. It was never trained on deepfakes, yet detects face swaps at 85%.

To investigate, we measure how each corruption type perturbs the constraint network's internal representations. For each of 100 paragraphs and each of 15 corruption types, we compute the displacement vector: the mean-pooled representation of the corrupted text minus the mean-pooled representation of the coherent original, taken from the layer immediately before the energy head. We then compute the cosine similarity between the mean displacement vectors across all corruption types (Appendix D, Table 9).

The result is striking: 13 of 15 corruption types produce displacement vectors with pairwise cosine similarity 0.80–0.98. The model has learned a dominant direction in representation space that corresponds to "deviation from coherence." Shuffled text, negated claims, swapped entities, shifted tenses, broken bridging references, and swapped role titles all push representations along this shared direction. We note that directional alignment of displacement vectors is a weaker property than the existence of a separating boundary—aligned perturbations could, in principle, exist without a clean decision surface. However, the independent evidence of 87.2% detection accuracy on unseen corruption types confirms that these perturbations do functionally cross a decision boundary in the energy landscape. The displacement analysis characterizes the geometry; the accuracy numbers confirm the boundary exists. Topic splice (0.51–0.79 similarity) and role/title swap (0.61–0.80) are partial outliers that deviate from the dominant direction, yet are still detected reliably (98% and 100% respectively), suggesting that "somewhat off the dominant direction" still crosses the decision surface.

The repetition corruption confirms this interpretation from the negative direction. Its displacement vectors have near-zero similarity with all other corruptions (0.10–0.24), meaning the representation barely moves. The model correctly identifies nothing structurally wrong: each repeated sentence is individually well-formed and present in the original paragraph. The one corruption that does not push along the "away from coherent" direction is the one corruption the model fails to detect (44%). This is not a deficiency in the experiment—it is the experiment working: the displacement analysis predicts both the successes and the failure.

Two secondary observations add detail. First, within the dominant direction, referential corruptions form the tightest sub-cluster: entity_swap $\leftrightarrow$ desc_swap = 0.981, entity_swap $\leftrightarrow$ name_sub = 0.956, desc_swap $\leftrightarrow$ bridging = 0.968. The model has some capacity to distinguish violation types, though this secondary structure is much weaker than the primary coherence direction. Second, topic splice is the most distinct non-repetition corruption (0.51–0.79 similarity with others), consistent with its qualitatively different nature: it replaces content entirely rather than perturbing it.

The encoder diagnostic provides the necessary condition for all of this: the encoder must preserve the relevant features (Section 5.3). The displacement analysis provides the geometric mechanism: corruptions that the encoder can distinguish from coherent text all push in a shared direction, enabling detection of unseen violations by the same boundary that catches trained ones.

The alpha ablation (Appendix E) provides one additional data point: the max term in the energy aggregator matters more as the test distribution moves further from training. In-distribution FF++ methods show $\pm 0.003$ AUC sensitivity to $\alpha$, while cross-dataset Celeb-DF shows 0.064 improvement from $\alpha = 0$ to $\alpha = 0.5$ (0.810 $\rightarrow$ 0.874). This is consistent with out-of-distribution violations producing sparser, more localized energy spikes that the max term captures—though it does not prove it.

This is the theoretical justification for corruption-as-specification: the corruption is not what the model detects. The structural property is. The corruption merely specifies which property to learn. The training signal is upstream of the test task—derived from a specification of structural coherence rather than from the test distribution's artifact statistics. A model trained on texture splice does not "know about" deepfakes; it knows about regional compatibility, and deepfakes happen to violate it.

### 5.3 The Encoder as Feature Space

The constraint network can only learn structure that exists in its input representations:

In text, MiniLM compresses shuffled and unshuffled paragraphs into similar vectors—discourse ordering violations become invisible. BERT preserves word order and entity identity—detection becomes possible. Switching encoders at the same data scale lifted accuracy from 54.5% to 93.4%.

In vision, a corruption alignment diagnostic showed DINOv2 encodes structural features well (texture boundaries, lighting gradients) but is blind to frequency-domain features (high-frequency texture loss, spectral power distribution). This motivated the multi-branch architecture: the structural branch detects what DINOv2 can see; the frequency branch processes a rendered heatmap view to detect what DINOv2 misses.

*Practical implication*: before designing corruption strategies, verify that the encoder preserves the relevant features. Embedding distance diagnostics provide a simple gate.

## 6 Representation Compatibility for Composable Constraints

Adding frequency detection as a composable second branch required extensive investigation. Five approaches failed—concatenating different feature types, parallel networks with different encoders, learned gating—all achieving less than 3% frequency contribution. The consistent failure: features from different representation spaces produce energies on incompatible scales, allowing one branch to dominate.

The resolution: both branches must process features from the same frozen encoder. Rendering frequency statistics as an RGB image and encoding through the same DINOv2 produced 44.4% frequency contribution—representation compatibility by construction. The detailed investigation timeline is provided in Appendix A.

This principle generalizes: any system composing multiple energy-based constraints must ensure the energies are on comparable scales. The simplest way is to process all views through the same encoder, ensuring the constraint networks operate in the same feature space.

## 7 Preliminary: Integration with Generative Models

As a preliminary validation that the constraint network can shape generative model training, we integrate it with Pythia-160m (Biderman et al., 2023)—a deliberately small model consistent with the paper's single-GPU efficiency constraint—by adding energy as a regularization term during LoRA fine-tuning:

$$\mathcal{L}_{\text{total}} = \mathcal{L}_{\text{LM}} + \lambda(t) \cdot E(\text{bridge}(\mathbf{h})) \tag{4}$$

where $\mathbf{h}$ are hidden states from a chosen layer, $\text{bridge}(\cdot)$ projects them into the constraint network's input space, and $\lambda(t)$ follows a sigmoid warmup schedule. The constrained model produces outputs with 1.64 lower energy than baseline at 0.05% perplexity cost.

Three technical challenges arose and were solved: bridge collapse (prevented by reconstruction loss that forces the bridge to faithfully encode LM hidden states), distribution mismatch (solved by aligning bridge outputs to pre-computed BERT embeddings), and co-adaptation instability (addressed by freeze-then-unfreeze with exponential moving average smoothing). These solutions may generalize to other settings where an external evaluator provides gradient signal to a generative model, but validation at larger scale and with ablations confirming that the bridge preserves energy ordering remain future work.

## 8 Limitations and Future Work

**Encoder ceiling**: DINOv2's $14 \times 14$ pixel patches cannot encode sub-patch texture variations—the granularity at which NeuralTextures and GAN smoothing artifacts occur. Upgrading to higher-resolution encoders requires changing only the input projection layer.

**Per-sample gating**: Dataset-level gating correctly enables/disables the frequency branch across datasets but cannot modulate per-image. Per-sample confidence estimation for composable constraints remains open.

**Repetition in text**: Verbatim repetition of an earlier sentence is detected at only 44%—locally coherent at the representation level, a genuine limitation of operating on embeddings.

**Absolute vs. paired scoring**: The model works reliably as a paired comparator but energy values are not fully comparable across different topics or domains.

**Generalization theory**: The displacement analysis (Section 5.2) shows that corruption types produce directionally aligned perturbations in representation space, consistent with the model learning a dominant coherence direction rather than category-specific detectors. The detection accuracy on unseen types (87.2%) independently confirms that these perturbations cross a functional decision boundary. However, in common with other corruption-based and contrastive methods, we lack a predictive theory of *which* unseen violations will be detected and at what accuracy. The encoder diagnostic provides a necessary condition; the displacement analysis provides geometric characterization; but a formal predictive model remains an open problem for self-supervised representation learning broadly.

**Benchmark coverage**: Vision evaluation uses FF++ and Celeb-DF. We have not tested on Celeb-DF++ (Li et al., 2025), the recently released harder benchmark where even SOTA detectors show ~7% AUC drops, nor on DFDC. Our contribution is the cross-modal architecture and corruption-as-specification framework; broader benchmark evaluation would strengthen the empirical case and is planned as future work.

**Future extensions**: The framework requires only new corruption strategies to extend to new domains— video (temporal coherence via inter-frame corruptions), code (type/scope violations), audio (spectral consistency). The composable branch architecture enables incremental addition without retraining existing branches. The displacement analysis (Section 5.2) suggests a predictive diagnostic for new domains: candidate corruption sets can be evaluated by their displacement vector geometry before full training, predicting whether they will produce a shared coherence direction or fragment into incompatible directions.

## 9 Related Work

**Discourse coherence**: Entity-grid models (Barzilay & Lapata, 2008) and neural approaches (Li & Hovy, 2014) require labeled data or explicit discourse relations. Our approach learns coherence from corruption-generated contrastive pairs.

**Deepfake detection**: Face X-ray (Li et al., 2020a) detects blending boundaries; RECCE (Cao et al., 2022) uses reconstruction-based methods. These methods are end-to-end trained on specific deepfake types and predate the foundation-model approach our work shares with (Yermakov et al., 2025; Cheng et al., 2025; Xia et al., 2026; Yan et al., 2025).

**Foundation-model deepfake detection**: Recent work has converged on frozen or minimally-adapted foundation encoders as the basis for generalizable detection. GenD (Yermakov et al., 2025) fine-tunes only the LayerNorm parameters of DINO/CLIP/PE encoders (0.03% of total) and reports state-of-the-art cross-dataset video-level AUROC across 14 benchmarks. Effort (Yan et al., 2025) adapts foundation models via orthogonal subspace decomposition. DFF-Adapter (Xia et al., 2026) inserts adapter heads into every DINOv2 transformer block. Independent analysis of frozen DINOv3 (Cheng et al., 2025) argues that frozen foundation models may already act as effective cross-generator detectors, and that large-scale adaptation can bias representations toward generator-specific artifacts. Our approach shares the frozen-encoder philosophy but differs structurally: we add no adaptation to the encoder, instead building a separate structural-reasoning module on its frozen patch features. This separation is what enables cross-modal applicability—the same constraint network architecture can be applied with a different encoder via a single projection layer change, while encoder-adaptation methods are tied to their specific encoder.

**Energy-based models**: Residual energy-based models for text (Deng et al., 2020) operate at the sequence level without per-position decomposition or cross-modal application.

**Contrastive learning**: SimCSE (Gao et al., 2021) uses generic noise for sentence embeddings. Our corruption strategies are structurally targeted—each breaks a specific property.

**Self-supervised vision**: DINOv2 (Oquab et al., 2024) provides the frozen patch features our vision branches operate on. Our contribution is the structural evaluation layer that detects coherence violations in these features.

## 10 Conclusion

Energy-based constraint networks learn structural coherence as a global distributional property of sequences, complementing the local conditional distributions that generative models capture. The architecture separates feature extraction (frozen encoders), structural reasoning (trainable constraint networks), and violation specification (corruption strategies), enabling cross-modal applicability with minimal parameters.

The key insight is that corruption-as-specification defines structural properties implicitly: you teach the system what structure to enforce by choosing what to break. The model then learns the underlying property, not the specific corruption—generalizing to violations it was never trained on in both text (87.2% on unseen types) and vision (0.850 AUC on unseen deepfake methods).

When combined with real-world paired data and composed with specialist branches, the system achieves 0.870 AUC on cross-dataset deepfake detection (mean over 5 seeds, ±0.019), competitive with systems using 2–3× more parameters and end-to-end training. The framework is encoder-agnostic and domain-agnostic—extending to new modalities requires only new corruption strategies and, optionally, paired data.

The broader implication is that structural coherence may be a more general and more portable property than typically assumed. The same architecture, trained with the same procedure, detects discourse violations in text and manipulation artifacts in images. Language models learn what comes next; this system learns what belongs together. The two are complementary, and their combination—a generator shaped by a structural prior—offers a principled alternative to post-hoc filtering or task-specific classifiers.

### Broader Impact Statement

This work presents a general framework for detecting structural violations in text and images. In deepfake detection, the technology supports media authenticity verification, which has positive societal implications. However, as with any detection technology, it could inform adversaries seeking to evade detection. We believe the benefits of open research in this area outweigh the risks, as detection methods must be publicly understood to be trusted. As with all current deepfake detection methods, false-negative behavior on unseen generation techniques is not characterized; no single-method detector should serve as a sole authenticity arbiter.

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

## A   Frequency Feature Investigation

Table 7: Approaches to incorporating frequency features. Only processing both views through the same frozen encoder achieved meaningful frequency contribution.

| Approach | Freq. contribution | Issue |
|---|---|---|
| SRM concat (808 dims) | ∼0% | Projection zeroed frequency columns |
| Dual network + SRM | <3% | Incompatible energy scales |
| Freeze-then-unfreeze | ∼10% (unstable) | Structural branch destabilized |
| Learned CNN encoder | ∼0% | Unstable gradient dynamics |
| Heatmap → small SSM | <1% | Too sparse for SSM |
| DINOv2(heatmap) → single net | — | SSM domain-switch failure |
| **DINOv2(heatmap) → separate net** | **44.4%** | **Compatible representations** |

## B   Experimental Configuration

Table 8: Configuration details for both domains.

| Parameter | Text | Vision |
|---|---|---|
| Encoder | BERT-base (110M, frozen) | DINOv2 ViT-B/14 (86M, frozen) |
| Input representation | 8-token windows | 16×16 patches |
| Positions per input | ≤32 | 256 |
| Embedding dim | 768 | 768 |
| Trainable params | 7.4M | 3.6M per branch |
| Training data | 50K Wikipedia paragraphs | FF++ (real/fake pairs) |
| Corruption types | 6 trained, 9 held out | 4 (texture, lighting, smoothing, blur) |
| Training time | ~45 min | ~30 min per branch |

## C  Concrete Text Detection Examples

The following examples show real model outputs. Per-position energy values are from the BERT-trained constraint network. Corrupted sentences are marked with ▶.

**Example 1: Tense shift (trained, $\Delta = +2.15$)**

**Coherent** ($E = +1.36$):

> [+1.45] Returning to Australia, Headlam became Deputy Chief of the Air Staff
>    (DCAS) on 26 January 1965.
> [+0.93] He was appointed a Companion of the Order of the Bath in
>    recognition of distinguished service in the Borneo Territories.
> [+1.13] His tenure as DCAS coincided with the most significant rearmament
>    program the Air Force had undertaken since World War II.

**Corrupted** ($E = +3.52$, $\Delta = +2.15$):

> [+2.13] Returning to Australia, Headlam became Deputy Chief of the Air Staff...
> [+1.54] He was appointed a Companion of the Order of the Bath...
> [+1.97] ▶ His tenure as DCAS coincided with the most significant rearmament
>    program the Air Force **has** undertaken since World War II.
> [+0.94] ▶ The first RAAF helicopters **are** committed to the Vietnam War...

**Example 2: Topic splice (trained, $\Delta = +2.14$)**

**Coherent** ($E = +1.51$): Military history paragraph about RAAF operations.

**Corrupted** ($E = +3.65$): Two middle sentences replaced with tropical storm text. Spliced sentences produce the highest per-position energies (+4.07, +3.96). Surrounding unchanged sentences also spike ($+1.26 \rightarrow +3.48$)—the violation destroys the structural context.

**Example 3: Role/title swap (unseen, $\Delta = +1.38$)**

This corruption was *never in training data*. "The captain noted that" is injected into text about a wing commander. The corrupted position scores +3.61 (highest in sequence), with disruption propagating to all surrounding positions. The model has learned that role references must be consistent with established context—a structural property it was never explicitly taught.

## D  Displacement Vector Analysis

For each corruption type, we compute the mean displacement vector in the constraint network's internal representation space: the average difference between the representation of corrupted text and the represen-

tation of its coherent original, taken from the layer immediately before the energy head. Table 9 shows the pairwise cosine similarity between these mean displacement vectors across all 15 corruption types.

Table 9: Cosine similarity between mean displacement vectors. [T] = trained, [U] = unseen. Most corruptions share a dominant "away from coherent" direction (0.80–0.98). Repetition is the clear outlier ($\sim$0.1), consistent with its 44% detection rate.

| | shuffle [T] | negate [T] | entity [T] | tense [T] | coref [T] | topic [T] | name [U] | desc [U] | demo [U] | sg/pl [U] | role [U] | temp [U] | repet [U] | num [U] | bridge [U] |
|---|---|---|---|---|---|---|---|---|---|---|---|---|---|---|---|
| shuffle | 1.00 | .95 | .95 | .95 | .92 | .74 | .95 | .95 | .89 | .93 | .70 | .84 | **.14** | .86 | .95 |
| negate | .95 | 1.00 | .93 | .92 | .90 | .69 | .93 | .93 | .89 | .88 | .72 | .86 | **.17** | .85 | .93 |
| entity | .95 | .93 | 1.00 | .94 | .90 | .75 | .96 | **.98** | .91 | .93 | .67 | .83 | **.23** | .83 | .95 |
| tense | .95 | .92 | .94 | 1.00 | .92 | .79 | .93 | .93 | .87 | .93 | .63 | .81 | **.22** | .81 | .92 |
| coref | .92 | .90 | .90 | .92 | 1.00 | .76 | .91 | .90 | .83 | .91 | .61 | .77 | **.17** | .77 | .89 |
| topic | .74 | .69 | .75 | .79 | .76 | 1.00 | .77 | .73 | .60 | .79 | .22 | .51 | **.24** | .51 | .69 |
| name | .95 | .93 | .96 | .93 | .91 | .77 | 1.00 | .96 | .89 | .92 | .67 | .82 | **.16** | .81 | .95 |
| desc | .95 | .93 | **.98** | .93 | .90 | .73 | .96 | 1.00 | .93 | .93 | .70 | .84 | **.15** | .82 | **.97** |
| demo | .89 | .89 | .91 | .87 | .83 | .60 | .89 | .93 | 1.00 | .86 | .80 | .88 | **.12** | .84 | .91 |
| sg/pl | .93 | .88 | .93 | .93 | .91 | .79 | .92 | .93 | .86 | 1.00 | .59 | .76 | **.16** | .76 | .90 |
| role | .70 | .72 | .67 | .63 | .61 | .22 | .67 | .70 | .80 | .59 | 1.00 | .82 | **-.03** | .76 | .72 |
| temp | .84 | .86 | .83 | .81 | .77 | .51 | .82 | .84 | .88 | .76 | .82 | 1.00 | **.12** | .87 | .84 |
| repet | **.14** | **.17** | **.23** | **.22** | **.17** | **.24** | **.16** | **.15** | **.12** | **.16** | **-.03** | **.12** | 1.00 | **.11** | **.10** |
| num | .86 | .85 | .83 | .81 | .77 | .51 | .81 | .82 | .84 | .76 | .76 | .87 | **.11** | 1.00 | .85 |
| bridge | .95 | .93 | .95 | .92 | .89 | .69 | .95 | **.97** | .91 | .90 | .72 | .84 | **.10** | .85 | 1.00 |

Three observations:

**Dominant coherence direction.** 13 of 15 corruption types produce displacement vectors with pairwise similarity 0.80–0.98. The model has learned a single dominant direction in representation space corresponding to "deviation from coherence." This directional alignment explains generalization: unseen corruptions push along the same direction that trained corruptions defined, and the independent detection accuracy (87.2% on unseen types) confirms these perturbations cross a functional decision boundary in the energy landscape.

**Repetition is geometrically absent.** Repetition's displacement vectors have near-zero similarity with all other corruption types (0.10–0.24). The representation barely moves because a paragraph with a repeated sentence contains the same individually well-formed sentences as the original. The 44% detection rate is predicted by this geometry: the corruption does not produce a perturbation along the dominant coherence direction.

**Secondary structure exists but is weak.** Within the dominant direction, referential corruptions form the tightest sub-cluster (entity $\leftrightarrow$ desc = 0.98, desc $\leftrightarrow$ bridge = 0.97). Topic splice is the most distinct non-repetition corruption (0.51–0.79), consistent with its qualitatively different nature of replacing content entirely rather than perturbing it. These secondary directions may encode violation type, but the primary geometric pattern is the shared coherence direction.

The analysis uses 100 paragraphs per corruption type. The qualitative pattern—dominant shared direction at 0.85+ versus repetition at 0.10–0.24—is stable across random subsamples of 50 paragraphs, with individual cosine values varying by $\pm$0.02–0.05.

# E Alpha Sensitivity Analysis

We ablate the energy aggregation coefficient $\alpha$ in $E(\mathbf{x}) = \text{mean}(e_i) + \alpha \cdot \text{max}(e_i)$ across both domains. In vision, we train all three branches with $\alpha \in \{0.0, 0.1, 0.2, 0.3, 0.5, 1.0\}$ at fixed seed. In text, we train the constraint network with the same $\alpha$ values. $\alpha = 0$ reduces to pure mean aggregation; higher values weight the worst-case position more heavily.

**Text domain:**

Four observations emerge across both modalities.

Table 10: Alpha ablation: combined three-branch AUC on vision benchmarks. The $\alpha = 0.3$ Celeb-DF value (0.863) is a single seed; the 0.870±0.019 reported in Table 4 is the mean over 5 seeds. The difference is within the reported variance.

| $\alpha$ | Deepfakes | Face2Face | FaceSwap | NT | FaceShifter | Celeb-DF |
|---|---|---|---|---|---|---|
| 0.0 | 0.956 | 0.910 | 0.922 | 0.849 | 0.892 | 0.810 |
| 0.1 | 0.957 | 0.909 | 0.917 | 0.869 | 0.895 | 0.793 |
| 0.2 | 0.958 | 0.911 | 0.918 | 0.876 | 0.897 | 0.842 |
| 0.3 | 0.956 | 0.904 | 0.916 | **0.888** | 0.897 | 0.863 |
| 0.5 | **0.962** | 0.910 | 0.921 | 0.874 | 0.897 | **0.874** |
| 1.0 | **0.962** | **0.913** | 0.918 | 0.879 | 0.897 | 0.858 |

Table 11: Alpha ablation: structural branch only (isolates the $\alpha$ effect from branch composition).

| $\alpha$ | Deepfakes | Face2Face | FaceSwap | NT | FaceShifter | Celeb-DF |
|---|---|---|---|---|---|---|
| 0.0 | 0.958 | 0.914 | 0.923 | 0.801 | 0.894 | 0.782 |
| 0.1 | 0.958 | 0.911 | 0.917 | 0.803 | 0.895 | 0.765 |
| 0.2 | 0.956 | 0.910 | 0.914 | 0.798 | 0.892 | 0.779 |
| 0.3 | 0.958 | 0.910 | 0.913 | 0.802 | 0.897 | 0.804 |
| 0.5 | 0.959 | 0.910 | 0.915 | 0.799 | 0.892 | 0.791 |
| 1.0 | 0.957 | 0.908 | 0.911 | 0.798 | 0.887 | 0.789 |

Table 12: Alpha ablation: text domain validation accuracy by corruption type. Tense shift (60%) and coreference break (65–67%) show lower raw accuracy than in Table 1 because this validation includes silent corruption failures (where the corruption returns unchanged text); Table 1 counts only pairs where the corruption actually modified the text, yielding 100% for both. The $\alpha$ sensitivity is zero for both types regardless of filtering.

| $\alpha$ | Overall | Shuffle | Negate | Entity | Tense | Coref | Topic |
|---|---|---|---|---|---|---|---|
| 0.0 | **0.792** | 86% | 94% | 82% | 60% | 67% | 86% |
| 0.1 | 0.785 | 86% | 93% | 81% | 60% | 65% | 86% |
| 0.2 | 0.786 | 86% | 93% | 81% | 60% | 66% | 86% |
| 0.3 | 0.787 | 86% | 93% | 81% | 60% | 66% | 86% |
| 0.5 | 0.784 | 85% | 93% | 81% | 60% | 66% | 86% |
| 1.0 | 0.770 | 82% | 93% | 80% | 60% | 65% | 84% |

First, $\alpha$ is insensitive in $[0.0, 0.5]$ in both domains. Text varies by ±0.004 (0.784–0.792); vision in-distribution varies by ±0.003 on FF++ Deepfakes. Only $\alpha = 1.0$ shows meaningful degradation in text ($-0.022$), suggesting the max term becomes dominant and overwhelms the mean signal at high values.

Second, the max term's contribution is most visible in cross-domain transfer. Vision cross-dataset Celeb-DF improves 0.064 from $\alpha = 0$ to $\alpha = 0.5$ (0.810 $\rightarrow$ 0.874). In text, the effect is negligible—consistent with the displacement analysis (Section 5.2), which showed text corruptions produce broad displacement vectors (0.80–0.98 cosine similarity) that the mean term already captures. Vision cross-dataset violations appear to produce sparser, more localized energy spikes where the max term adds information.

Third, per-type sensitivity in text is minimal: shuffle, negate, tense shift, and topic splice are essentially unchanged across $\alpha \in [0.0, 0.5]$. The structural branch in vision (Table 11) shows similar stability, varying by ±0.001 on Deepfakes.

Fourth, the value $\alpha = 0.3$, selected in the text domain and transferred to vision without retuning, falls in the stable region for both modalities. Any value in $[0.2, 0.5]$ would produce comparable results in both domains,

confirming that the qualitative role of the max term matters more than its precise weight, and that a single hyperparameter value applies across modalities without adjustment.

## F  Beta Sensitivity Analysis: Branch Combination Weight

We ablate the local texture branch weight $\beta$ in $E = E_{\text{structural}} + \text{gate} \cdot E_{\text{frequency}} + \beta \cdot E_{\text{local}}$ across $\beta \in \{0.0, 0.1, 0.2, 0.3, 0.5, 0.7, 1.0\}$. $\beta = 0$ reduces to structural-only (frequency gate is disabled on this dataset); $\beta = 1$ gives equal weight to structural and local branches.

Table 13: Beta ablation: combined AUC on FF++ methods. $\beta = 0.3$ (used in all reported results) balances coverage across manipulation types.

| $\beta$ | Deepfakes | Face2Face | FaceSwap | NT | FaceShifter | $\Delta$NT |
|---|---|---|---|---|---|---|
| 0.0 | 0.959 | 0.918 | 0.917 | 0.819 | 0.908 | — |
| 0.1 | 0.962 | 0.923 | 0.924 | 0.862 | 0.912 | +0.043 |
| 0.2 | 0.962 | 0.923 | 0.924 | 0.873 | 0.912 | +0.054 |
| **0.3** | **0.962** | **0.922** | **0.924** | **0.882** | **0.912** | **+0.063** |
| 0.5 | 0.960 | 0.919 | 0.921 | 0.895 | 0.910 | +0.076 |
| 0.7 | 0.958 | 0.915 | 0.918 | 0.903 | 0.906 | +0.084 |
| 1.0 | 0.953 | 0.907 | 0.911 | 0.911 | 0.900 | +0.092 |

Three observations. First, the local branch contributes substantially: NeuralTextures improves from 0.819 ($\beta=0$, structural-only) to 0.882 at $\beta=0.3$, a +0.063 gain. This demonstrates that composition adds value— the combined system detects manipulation types that no single branch handles well. Second, there is a clear tradeoff: higher $\beta$ helps NeuralTextures (local texture violations) but hurts structural-dominant methods (Deepfakes drops from 0.962 to 0.953 at $\beta = 1.0$). The value $\beta = 0.3$ provides the best coverage across unknown manipulation types, gaining +0.063 on NeuralTextures while losing only $-0.003$ on Deepfakes. Third, any $\beta \in [0.1, 0.5]$ produces good overall results; the system is not sensitive to the precise combination weight.

