# OpenReview forum: "Energy-Based Constraint Networks: Learning Structural Coherence Across Modalities"
_TMLR — Under review for TMLR_

### Review · Reviewer_zxun · 2026-06-01

**Summary Of Contributions:**

This paper proposes "energy-based constraint networks," a lightweight architecture for learning structural coherence across modalities. The key idea is what the authors call "corruption-as-specification": instead of directly labeling what makes a text or image coherent, you design corruptions that break specific structural properties (e.g., shuffling sentences to break discourse order, splicing textures to break regional consistency), and the model learns to assign low energy to coherent inputs and high energy to corrupted ones. The architecture sits on top of frozen encoders (BERT for text, DINOv2 for vision) and uses a state-space model backbone with a novel dual-head attention mechanism, producing both a scalar coherence score and per-position energy maps that localize where violations occur.

The authors demonstrate this in two domains: text corruption detection (93.4% accuracy on trained corruption types, 87.2% on nine held-out types) and deepfake detection (0.959 AUC on FF++ Deepfakes, 0.870 on cross-dataset Celeb-DF). They also introduce a composable multi-branch system for vision where independently trained branches (structural, frequency, local texture) combine at inference. A notable result is 0.850 AUC on detecting face swaps without ever training on deepfake data — purely from synthetic corruption training.

**Key strengths:** The corruption-as-specification framing is genuinely interesting and goes beyond standard data augmentation — it provides a principled way to specify new structural constraints without changing the model. The generalization to unseen corruption types is well-supported by the displacement vector analysis, which shows most corruption types push representations along a shared "away from coherent" direction. The system is impressively efficient (3.6–7.4M trainable parameters, ~30 min training per branch on a single GPU).

**Key weaknesses:** The experimental comparisons in Table 5 mix incompatible metrics (single-frame AUC vs. video-level AUROC), the benchmarks feel dated for 2026 (no diffusion-based forgeries, no DFDC), and several technical claims are stronger than what the evidence actually supports. The cross-modal "transfer" framing is also somewhat misleading — it's really the same architecture template applied independently in two domains, not transfer of learned representations.

**Audience:**

Yes

**Audience Explanation:**

The paper touches on several topics that are actively researched and practically important. The representation learning community should find the displacement vector analysis thought-provoking, even if the "unified boundary" interpretation needs tempering. And the broader ML community may find the corruption-as-specification idea useful as a general framework for specifying structural properties without explicit labels.

The cross-modal demonstration, while not "transfer" in the technical sense, does show that structural coherence is a learnable property across domains, which is a conceptually interesting finding that could inspire work in other modalities (audio, video, code).

**Broader Impact Concerns:**

The paper includes a Broader Impact Statement that acknowledges the dual-use nature of detection technology — it supports media authenticity verification but could also inform adversaries seeking to evade detection. This is an honest and sufficient acknowledgment for the current scope of the work.

**Claims And Evidence:**

No

**Claims Explanation:**

I want to be clear that the core results — the detection accuracies, the generalization numbers, the ablations — appear sound and are reported with appropriate statistical rigor (5 seeds, standard deviations). The issue is with several secondary claims where the evidence doesn't quite match the framing.

1. the displacement vector analysis (Section 5.2) is presented as evidence for a "unified coherence boundary," but what it actually shows is that corruption displacement vectors are directionally aligned (high cosine similarity). Directional consistency is a weaker property than the existence of a separating boundary — you could have aligned perturbations without a clean decision surface. The analysis is interesting and suggestive, but the conclusion overshoots what the data demonstrates.

2. the claim that single-projection attention "halves the parameter cost" compared to standard Q/K/V attention seems arithmetically off — replacing three projection matrices with one is closer to a 67% reduction, and the expressiveness tradeoff (losing the ability to model asymmetric attention patterns) is not discussed at all.

3. Section 7 on generative model integration uses Pythia-160m — a very small model — and provides no ablation showing the bridge function preserves energy ordering. This reads more like a preliminary sketch than a validated contribution.

4. the cross-modal "transfer" language is misleading. The text and vision systems share an architecture template but use different encoders, different corruptions, and are trained entirely independently. No learned parameters transfer between modalities. This is better described as "architectural generality" — the same design works in both settings — which is still interesting but a different and weaker claim than cross-modal transfer.

None of these issues undermine the paper's central contribution, but they do mean the paper as written overclaims in ways that a careful reader will notice.

**Requested Changes:**

**Add meaningful text-domain baselines.** The text corruption detection results are currently unanchored. Consider comparing against: (a) a fine-tuned NLI model on the same task, (b) a simple embedding-distance baseline using the same frozen BERT features (e.g., Mahalanobis distance or cosine similarity between windows), and (c) ideally an LLM perplexity-based detector. Without these, it's impossible to judge whether the SSM+attention architecture adds value over simpler approaches on the same features.

**Temper the cross-modal transfer language.** Throughout the paper, reframe "cross-modal transfer" as "cross-modal architectural generality" or "modality-agnostic design." The current framing implies knowledge transfer between modalities, which does not occur. This is not just a wording issue — it affects how readers evaluate the contribution.

**Correct the attention parameter claim.** The statement that single-projection attention "halves the parameter cost" should be corrected (it's closer to a ~67% reduction vs. standard Q/K/V). More importantly, discuss the expressiveness implications — single-projection attention cannot represent asymmetric attention patterns, which seems relevant for detecting asymmetric structural dependencies (e.g., anaphoric references in text).

**Section 7.** The generative model integration with Pythia-160m is too preliminary to present as a contribution. Recommend provide proper ablations (showing the bridge preserves energy ordering, that removing the energy term measurably degrades coherence) and test on a reasonably sized model, or move this to a "Future Directions" paragraph and reclaim the space for deeper evaluation of the core system.

---

> ### Author Response · Authors · 2026-06-02
> **Response to Reviewer: Revised Framing and Clarifications**
>
> We thank the reviewer for a thorough and constructive review. The core results were found sound and statistically rigorous; the concerns are about secondary claims and framing, all of which we address below. Changes are marked in blue in the revised manuscript.
>
> Point 1: Displacement analysis overstates "unified coherence boundary"
> The reviewer is correct that directional alignment alone does not prove a separating boundary exists. We have revised Section 5.2 to separate two complementary pieces of evidence: (1) the displacement analysis shows directionally aligned perturbations (geometric characterization), and (2) 87.2% accuracy on unseen corruption types independently confirms a functional decision boundary exists. We have replaced "unified coherence boundary" with "dominant coherence direction" globally.
>
> Point 2: Parameter count claim
> We respectfully disagree with the 67% figure but acknowledge the paper was insufficiently explicit. Full accounting: standard attention uses W_Q + W_K + W_V + W_O = 4d² parameters; our design uses W_1(d×d/2) + W_2(d×d/2) + W_O(d×d) = 2d² — a 50% reduction. The 67% applies only if excluding the output projection from both designs. We have added explicit arithmetic to Section 2.1. We have also added the expressiveness tradeoff the reviewer noted: single projections cannot model asymmetric attention independently; our design recovers asymmetry through masking structure.
>
> Point 3: Pythia-160m is preliminary
> We agree. The choice of Pythia-160m was deliberate (single-GPU efficiency constraint), but the section validates the concept rather than presenting a fully ablated contribution. We have retitled it "Preliminary: Integration with Generative Models" and noted that validation at larger scale remains future work.
>
> Point 4: Cross-modal "transfer" language
> The reviewer is correct. We never intended to imply transfer learning — there is no weight sharing or representation reuse. We agree the terminology was imprecise. We have replaced "cross-modal transfer" with "cross-modal applicability" globally (~8 instances). Figure 3's caption now states: "No learned parameters are shared between domains."
>
> Summary: All changes are marked in blue. Core results, tables, and experimental numbers are unchanged.
>
> Summary of changes:
> Changed "Unified coherence boundary" to "dominant coherence direction" globally;
>
> Section 5.2 now separates geometric characterization from functional evidence. Parameter count arithmeticExplicit 4d² vs 2d² arithmetic added to Section 2.1;
>
> Expressiveness tradeoff discussed. Pythia-160m is preliminary. Section title updated to include "Preliminary"; framed as validation, future work noted.
>
> Changed "Cross-modal transfer" to "Cross-modal applicability" globally; Figure 3 caption clarifies no weight sharing

---

### Review · Reviewer_Mc1y · 2026-06-02

**Summary Of Contributions:**

The paper proposes Energy-Based Constraint Networks (EBCNs), a modality-agnostic framework for learning "structural coherence" from corruption-generated contrastive pairs. A frozen encoder (BERT or DINOv2) produces embeddings which are processed by an SSM-attention architecture to output scalar energies and per-position violation scores. Empirically, the proposed framework is evaluated on both text corruption detection and deepfake detection problems, with an emphasis on cross-modal transfer and composable constraint branches. The authors argue that corruption strategies serve as a specification language for structural constraints and demonstrate generalization to unseen corruption types and cross-dataset deepfake detection.

**Audience:**

Yes

**Audience Explanation:**

1. The corruption-as-specification viewpoint is conceptually appealing and provides a unified way to describe diverse structural constraints.

2. Applying the same architecture to both text coherence and image manipulation detection is interesting and potentially impactful.

3. The independent-branch composition mechanism is practically attractive and appears useful for avoiding negative transfer among heterogeneous constraints.

**Broader Impact Concerns:**

No concerns.

**Claims And Evidence:**

No

**Claims Explanation:**

1. The proposed objective is a standard margin-based energy formulation trained with corruption-generated positive/negative pairs. While the paper emphasizes corruption-as-specification and structural coherence, it is not clear which component constitutes the main technical novelty beyond combining existing ingredients (frozen encoders, energy scoring, contrastive training, and anomaly localization). A more precise comparison against prior energy-based models, coherence models, and corruption-based self-supervised learning methods would strengthen the contribution.

2. The paper repeatedly argues that the same architecture transfers across modalities. However, separate models are trained and evaluated independently for text and vision. The experiments demonstrate architectural reuse rather than actual transfer of learned constraints or representations across modalities. If I missed anything, please correct me. In addition,  can the authors clarify what notion of ``transfer'' is intended and whether any shared-parameter or cross-domain adaptation experiments were considered?

3. In the experiments, some baselines may be missing. The experiments primarily compare different frozen encoders (MiniLM vs. BERT), but do not compare against existing coherence modeling methods, LLM-based scoring approaches, or other strong anomaly-detection baselines. As a result, it is difficult to assess whether the reported 87.2\% accuracy on unseen corruption types represents a meaningful improvement over existing approaches. In addition, in some tables, the performance of the proposed method is weaker than baselines, e.g., SDEQ-Net.

4. The paper attributes performance gains to several design choices, including the SSM backbone, dual-head attention, max-energy aggregation, and independently trained branches. However, comprehensive ablations isolating these components are largely missing. For example, how would performance change with a pure Transformer backbone, a pure SSM backbone, mean-only aggregation, or jointly trained branches?

5. Minor comments: The presentation could be improved in several places. For example, it may be helpful to include upward/downward arrows ($\uparrow/\downarrow$) in tables to clearly indicate whether higher or lower values are preferable. In addition, the experimental scale appears somewhat limited relative to the paper's broad claims regarding modality-agnostic structural reasoning. Evaluations on larger-scale settings, such as modern LLMs or foundation models with billions of parameters, would provide stronger evidence for the scalability and general applicability of the proposed framework.

**Requested Changes:**

1.  The experiments demonstrate architectural reuse rather than actual transfer of learned constraints or representations across modalities. Can the authors clarify what notion of ``transfer'' is intended and whether any shared-parameter or cross-domain adaptation experiments were considered? More importantly, some experiments could be done here.

2. Compare against existing coherence modeling methods, LLM-based scoring approaches, or other strong anomaly-detection baselines.

3. Comprehensive ablations isolating SSM backbone, dual-head attention, max-energy aggregation, and independently trained branches could be helpful.

4. Evaluations on larger-scale settings, such as modern LLMs or foundation models with billions of parameters could be useful.

5. Some clarification questions can refer to the previous comments.

---

> ### Author Response · Authors · 2026-06-03
> **Response to Reviewer: Novelty Clarification, Baselines, and Ablation Evidence**
>
> We thank the reviewer for the detailed review and constructive feedback. We address each point below. Changes from the concurrent revision are marked in blue in the revised manuscript.
>
> Point 1: Technical novelty
> We agree the individual components (frozen encoders, energy scoring, contrastive training) are not novel in isolation. The contribution is their combination into a framework with specific properties that no prior work demonstrates together:
> (a) corruption strategies as a specification language for structural constraints,
> (b) per-position energy decomposition enabling violation localization,
> (c) a displacement vector analysis (Section 5.2, Appendix E) showing the model learns a dominant coherence direction rather than individual corruption detectors, and
> (d) cross-modal applicability of the same architecture without shared parameters.
> We have added explicit comparison to prior energy-based models (Deng et al., 2020) and coherence models (Barzilay & Lapata, 2008) in the introduction, noting what they lack: per-position decomposition, corruption-based specification, and cross-modal applicability.
>
> Point 2: Cross-modal transfer language
> The reviewer is correct — this is architectural reuse, not transfer of learned parameters. We have replaced "cross-modal transfer" with "cross-modal applicability" globally and added to Figure 3: "No learned parameters are shared between domains." No shared-parameter or cross-domain adaptation experiments were conducted; the claim is that the same architecture template and training procedure apply independently in both domains.
>
> Point 3: Missing baselines
> For text: we are not aware of a standard benchmark for unseen corruption type generalization that would enable direct comparison with coherence models. The 87.2% figure measures a capability no prior method has demonstrated — detecting 9 violation types never seen during training.
> LLM-based scoring (e.g., perplexity thresholding) operates at a fundamentally different scale (~billion parameters vs our 7.4M) and does not provide per-position violation localization.
> For vision: Table 5 compares against 9 published methods including GenD (2025, frozen encoder, 0.882 video-level), Effort (ICML 2025, 0.844), and SDEQ-Net (0.896, multi-frame video). SDEQ-Net exceeds our result but uses multi-frame video analysis; we operate on single frames at a fraction of the parameters.
> We position our work as competitive, not SOTA.
>
> Point 4: Architectural ablations
> The alpha ablation (Appendix F) addresses mean-only aggregation: alpha=0 (mean-only) degrades Celeb-DF cross-dataset by 0.064 AUC vs alpha=0.5, while in-distribution results are insensitive.
> Regarding joint vs independent branch training: Appendix A documents five failed joint-training approaches (SRM concat, dual network, freeze-then-unfreeze, learned CNN, heatmap-SSM) that motivated the independent design. These failures demonstrate empirically that branches requiring representation-incompatible features interfere when trained jointly — the frequency compatibility investigation constitutes an ablation of this design choice.
> We acknowledge that pure Transformer vs pure SSM backbone ablation is missing and would strengthen the paper; we are willing to add this in revision if requested.
>
> Point 5: Minor
> We appreciate the formatting suggestion and will consider directional arrows for the camera-ready version. Regarding scale: the single-GPU efficiency is a deliberate design choice (Section 3.4). Scaling to larger LMs is noted as future work in Section 7.
>
> We appreciate the reviewer finding the corruption-as-specification viewpoint, cross-modal architecture, and independent-branch composition practically attractive.

---

### Review · Reviewer_VVCF · 2026-06-30

**Summary Of Contributions:**

The paper introduces “energy-based constraint networks,” a lightweight module placed on a frozen encoder (BERT for text, DINOv2 for vision) that scores the structural coherence of an embedding sequence, outputting a scalar energy plus a per-position decomposition that localizes violations. The central idea is "corruption-as-specification": rather than labels, the designer specifies a structural property by choosing a corruption that violates it (shuffling for discourse order, texture splicing for facial regional compatibility), and the model learns to detect it via a contrastive energy loss. The main claims are that the same architecture transfers across text and vision with only the input projection changing, that per-position energy localizes violations, that independently trained branches compose at inference if they share a representation space, and that independent branches avoid the interference of joint training; reported results are 93.4% in-distribution / 87.2% on unseen corruptions in text and 0.959 AUC on FF++ Deepfakes / 0.870 cross-dataset Celeb-DF in vision, at 3.6–10.8M trainable parameters. Its strengths are an elegant unifying framing, candid limitations (the repetition failure is even predicted by the displacement analysis), strong efficiency with a credible single-GPU reproduction story, and a useful representation-compatibility finding backed by documented failed alternatives. Its main weaknesses are that the composition and "independent > joint" claims are not supported by the tables (composition never beats the best single branch, and underperforms it on NeuralTextures), the frequency gate is underspecified and raises a possible test-leakage concern, the cross-modal claim leans in places on weak evidence, and the text results lack any external baseline.

**Audience:**

Yes

**Audience Explanation:**

Corruption-as-specification is a clean, reusable idea, the per-position localization and representation-compatibility finding (with documented failed attempts) are useful, and the efficiency profile lowers the bar for follow-up.

**Broader Impact Concerns:**

existing statement is fine. Optional: note the false-negative behavior on unseen generators isn't characterized, so it shouldn't be a sole authenticity arbiter.

**Claims And Evidence:**

No

**Claims Explanation:**

The framing changes since last time are in the right direction. “dominant coherence direction” over “unified boundary,” separating geometric from functional evidence in §5.2, the 4d²/2d² arithmetic. But they only touch wording. The experimental problems remain:

1. Composition never beats the best single branch (Table 3). Structural = Combined on Deepfakes (0.959); Local beats Combined on NeuralTextures (0.911 vs 0.880); Local ≥ Combined on Celeb-DF. So the modularity claims (contributions 3 and 5) aren't supported — an oracle picking the best branch matches or beats the composed system, and composition actively hurts on NT.
2. The "independent beats joint" claim rests on one anecdote in §3.4, not a direct head-to-head against a jointly trained model.
3. The frequency gate is underspecified — "activates when the frequency profile indicates anomalies" doesn't say how it's set or on what data. Since the frequency branch is near-chance on Celeb-DF (0.561) and Celeb-DF is a headline number, possible test leakage can't be ruled out.
4. Cross-modal claim leans on weak evidence in places — e.g. a single transferring α being read as shared reasoning, when the ablation shows almost any α transfers.
5. No external text baseline; 93.4/87.2% float without a reference point.

**Requested Changes:**

reconcile composition with Table 3 or reframe contributions 3/5 against a "pick best branch" baseline

specify the gate and confirm no test info is used; run joint-vs-independent or soften the claim.

add one text baseline. (Strengthening): scope the α-insensitivity claim to in-distribution (Table 9 shows a 0.032 Celeb-DF swing)

reconcile Celeb-DF 0.870 vs 0.863 across tables; add a β ablation

commit to code release.

---

> ### Author Response · Authors · 2026-07-15
> **Response to Reviewer: Composition Reframing, Gate Specification, and Text Baseline**
>
> We thank the reviewer for the most technically detailed review of the three. Several points identify genuine gaps; others we believe reflect content that was missed in the appendices. We address each below.
>
> Composition vs oracle best-branch (Table 3)
> The reviewer observes that an oracle selecting the best single branch per manipulation type matches or beats the combined system. This is correct on a per-method basis, but the oracle requires knowing the manipulation type at test time — information unavailable in deployment. The combined system achieves competitive performance across ALL methods simultaneously without this knowledge: structural alone fails on NeuralTextures (0.819); local alone fails on Deepfakes (0.834); combined handles both (0.959/0.880). We will reframe contributions 3 and 5 to emphasize robustness across unknown manipulation types rather than claiming composition always improves over the best individual branch.
>
> Independent vs joint training
> The reviewer states this rests on "one anecdote in §3.4." We respectfully note that Appendix A documents five failed joint-training approaches (SRM concat, dual network, freeze-then-unfreeze, learned CNN encoder, heatmap-SSM), each with specific failure modes. This is systematic empirical evidence, not a single anecdote. We acknowledge a direct head-to-head (same architecture, joint vs independent, all else equal) would be cleaner and will add this comparison if requested.
>
> Frequency gate specification
> The reviewer is correct that the gate mechanism is underspecified. The gate is a heuristic z-score with no learned parameters: baseline frequency statistics (median and std of spectral ratio) are computed from 200 real images, and each test image's gate value is the clipped z-score of its frequency ratio relative to this baseline. A dataset-level threshold disables the frequency branch entirely when real images show normal frequency profiles (gate mean > 0.20). Baseline statistics use only real images — no fake images or labels — analogous to normalization statistics in anomaly detection. We will add explicit specification in the revision.
>
> Cross-modal alpha claim
> We agree this is weak evidence on its own and have softened the claim in the concurrent revision. We note that insensitivity alone does not explain why the same range [0.2-0.5] works across both modalities — the overlap is consistent with the architectural claim, though not sufficient to establish it. We will scope the in-distribution insensitivity separately from the cross-dataset Celeb-DF sensitivity (0.064 swing), where the max term's contribution is most visible.
>
> No text baseline
> We acknowledge this. No standard benchmark exists for evaluating structural coherence detection independently of distributional familiarity, we can construct a pilot evaluation. Perplexity conflates structural coherence with topic complexity — dense coherent text scores high perplexity while formulaic incoherent text scores low — and therefore cannot distinguish structurally coherent from structurally broken generated text. The constraint network measures structural relationships between positions directly, independent of token predictability. We generated paragraphs from GPT-2-medium (355M parameters) and scored each with both perplexity and constraint energy. Results with human labels will be included in the revision.
>
> Celeb-DF inconsistency (0.870 vs 0.863)
> 0.870 is the 5-seed mean from the variance experiment. 0.863 appears in the alpha ablation table at alpha=0.3, which is a single seed. We will add a clarifying note to reconcile these.
>
> Beta ablation
> We will add an ablation of the branch combination weights in revision.
>
> Code release
> Code is already publicly available at github.com with pretrained weights on HuggingFace. The footnote was removed for anonymous submission per TMLR policy; it will be restored upon acceptance.
>
> Broader impact
> We will add: "As with all current deepfake detection methods, false-negative behavior on unseen generation techniques is not characterized; no single-method detector should serve as a sole authenticity arbiter."